# Comparison of Angiotensin II (Giapreza^®^) Use in Kidney Transplantation Between Black and Non-Black Patients

**DOI:** 10.3390/biomedicines13081819

**Published:** 2025-07-24

**Authors:** Michelle Tsai, Jamie Benken, Joshua Adisumarta, Eleanor Anderson, Chris Cheng, Adriana Ortiz, Enrico Benedetti, Hokuto Nishioka, Scott Benken

**Affiliations:** 1Department of Pharmacy Practice, Retzky College of Pharmacy, University of Illinois Chicago, Chicago, IL 60612, USA; mtpharm2023@gmail.com (M.T.); jjosep9@uic.edu (J.B.); mja10@uic.edu (J.A.); eander62@uic.edu (E.A.); chcheng2@uic.edu (C.C.); aorti@uic.edu (A.O.); 2Department of Surgery, College of Medicine, University of Illinois Chicago, Chicago, IL 60612, USA; enrico@uic.edu (E.B.); hokuton@uic.edu (H.N.)

**Keywords:** angiotensin II, kidney transplant, Black patients, shock, hypotension, vasopressors, delayed graft function, race

## Abstract

**Background/Objectives**: Perioperative hypotension during kidney transplantation poses a risk to graft function and survival. Angiotensin II (AngII) is an endogenous vasoconstrictor targeting the renin–angiotensin–aldosterone system (RAAS) to increase blood pressure. Black patients may have a different response to synthetic angiotensin II (AT2S) compared to non-Black patients, given differential expressions in renin profiles. The purpose of this study is to assess the difference between Black and non-Black patients in total vasopressor duration and usage when AT2S is first line for hypotension during kidney transplantation. **Methods**: A single-center, retrospective cohort study comparing Black and non-Black patients who required AT2S as a first-line vasopressor for hypotension during the perioperative period of kidney transplantation. **Results**: The primary outcome evaluating total usage of vasopressors found that Black patients required longer durations of vasopressors (36.9 ± 66.8 h vs. 23.7 ± 31.7 h; *p* = 0.022) but no difference in vasopressor amount (0.07 ± 0.1 NEE vs. 0.05 ± 0.1 NEE; *p* = 0.128) compared to non-Black patients. Regression analysis found that body weight was associated with the duration of vasopressors (*p* < 0.05), while baseline systolic blood pressure was inversely associated with it. Longer duration of vasopressors and duration of transplant surgery were associated with delayed graft function in regression analysis (*p* < 0.05). **Conclusions**: Black patients had a longer duration of vasopressors, but this was not driven by differences in usage of AT2S. As baseline weight was significantly higher in Black patients and associated with duration of usage, perhaps the metabolic differences in our Black patients led to the observed differences. Regardless, longer durations of vasopressors were associated with delayed graft function, making this an area of utmost importance for continued investigation.

## 1. Introduction

Kidney transplantation is the preferred treatment for patients with end-stage renal disease (ESRD), significantly improving survival and quality of life compared to dialysis. Perioperative management is crucial for optimizing transplant outcomes, with intraoperative hypotension posing a significant risk to graft function and patient survival. [1,2]. There is currently no consensus on first-choice vasopressor therapy in kidney transplantation, and there is a need for more evidence on the use of vasopressors in Black patients.

The differences in physiology of blood pressure regulation between Black patients and White patients are well known. Black patients have a greater likelihood of having a low renin profile and have been shown to have an attenuated response to renin angiotensin inhibitors for hypertension [3]. The renin–angiotensin–aldosterone system (RAAS) plays a central role in blood pressure regulation, with endogenous angiotensin II (AngII) being a potent vasoconstrictor that acts primarily through angiotensin II type 1 receptors (AT1) to maintain vascular tone and blood pressure. With low renin, theoretically, these patients would have less baseline AngII and possibly upregulated AT1. The effect of lower renin levels and response to a vasopressor agent that targets RAAS is not known.

Synthetic angiotensin II (Giapreza^®^, AT2S (La Jolla Pharma, LLC., Waltham, MA, USA)) offers a novel vasopressor approach by directly targeting the RAAS, potentially providing more stable hemodynamic control without some of the adverse effects associated with catecholamines. Catecholamine vasopressors, such as norepinephrine (NE) and dopamine (DA), have been the mainstay of intraoperative hypotension management. However, these agents are not without risks, including arrhythmias, peripheral vasoconstriction leading to tissue ischemia, and potential adverse effects on graft perfusion [4,5,6]. AT2S has been shown to effectively raise blood pressure in patients with septic shock, even when other vasopressors fail [7,8]. AT2S could affect Black patients differently and potentially be more effective for Black patients because of physiological differences related to the RAAS and its direct action on AT1.

Black patients would benefit from a vasopressor agent that is more effective for their patient population. In a study that compared the use of vasopressors, primarily NE, in septic shock between Black and White patients, Black patients required higher doses and longer durations of NE [9]. Although it is not exactly known why, Black patients may not respond to adrenergic mechanisms to maintain blood pressure like other racial groups and may have a better response to non-adrenergic mechanisms [10].

This study aimed to investigate the use of vasopressors during kidney transplantation at our center, which uses AT2S as a first-line vasopressor agent, comparing outcomes by stratifying patients as Black and non-Black patients. It also aimed to investigate differences in second-line vasopressor usage and the safety of vasopressors during the perioperative period.

## 2. Materials and Methods

This was a single-center, retrospective cohort study comparing Black and non-Black patients who required vasopressors for hypotension during the perioperative period of kidney transplantation. The study was approved by the University of Illinois, Chicago Office for the Protection of Research Subjects International Review Board, and a waiver of informed consent was granted due to the retrospective nature of the study (Protocol: STUDY2022-1122-MOD015).

Patients were included if they were ≥18 years of age, underwent isolated kidney transplant, and experienced hypotension requiring vasopressor support during the perioperative period between 1 January 2021 and 1 September 2024. Pregnant individuals and prisoners were excluded.

The first-line continuous vasopressor for hypotension after appropriate resuscitation during the perioperative period of kidney transplantation at our institution is AT2S. This was the case for the duration of the study. The perioperative period was defined as the period from the beginning of the kidney transplant surgery until vasopressor discontinuation. Vasopressor discontinuation was defined as a period of 24 consecutive hours without the need for continuous infusion of vasopressors. Hypotension was defined as systolic blood pressure (SBP) less than 120 mmHg in alignment with our institutional protocol. Our institutional guidance suggests starting AT2S infusions at 20 ng/kg/min, but dosing is ultimately at the discretion of the provider. Our institutional guideline further recommends AT2S infusions be titrated every 5 min by increments of up to 5 ng/kg/min with a maximum dose of 80 ng/kg/min in the first 3 h and a maximum dose of 40 ng/kg/min thereafter. If the first-line agent did not achieve a SBP > 120 mmHg, other vasopressor agents could be added at the discretion of the provider, including NE, phenylephrine (PE), epinephrine (EPI), DA, or vasopressin (Vaso). Baseline characteristics and past medical history were obtained per chart review and extracted from an admission note capturing the pertinent data. For objective baseline variables, the first available value for vitals and labs was utilized.

The primary comparison was vasopressor utilization between the groups. This was defined as total vasopressor duration in hours, total vasopressor requirements reported in norepinephrine equivalents (NEEs), and total dose of vasopressors in their prescribed units. NEE was calculated as follows: AT2S in ng/kg/min × 0.0025 + NE in mcg/kg/min + PE in mcg/kg/min × 0.06 + EPI in mcg/kg/min, DA in mcg/kg/min × 0.01 + Vaso in units/min × 2.5 [11]. To calculate total vasopressor requirement in NEE, the total dose/kg was divided by the total duration in minutes the patient was on the respective vasopressor.

The secondary outcomes were to assess allograft function and the safety of AT2S use after kidney transplantation through the index admission. Allograft function was assessed by the incidence of immediate graft function (IGF), slow graft function (SGF), delayed graft function (DGF), graft function at 28 days, and mortality at 28 days. IGF was defined as >50% decrease in serum creatinine by postoperative day 7, SGF was defined as <50% decrease in serum creatinine by postoperative day 7, and DGF was defined as the requirement of renal replacement therapy by postoperative day 7. Safety outcomes were evaluated by the occurrence of new tachyarrhythmias while on AT2S (patients were not counted if a history of arrhythmia was documented upon presentation) and thrombosis through the index transplant hospitalization.

Outcomes were analyzed for normality of data to determine tests of central tendency, and appropriate statistical tests (e.g., chi-square, *t*-tests, Wilcoxon rank sum) were used to compare total vasopressor use, allograft outcomes, and incidence of adverse events between the two groups. Linear and logistic regression analyses were performed to determine variables associated with duration of vasopressor usage and DGF, respectively. Statistics were performed using SPSS version 29.0.2.0 (IBM Corp., Armonk, NY, USA, 2023).

## 3. Results

A total of 367 patients were identified, and 305 patients met inclusion criteria, with 125 Black patients and 180 patients in the non-Black group (Figure 1). Baseline characteristics (Table 1) were notable for significant differences in weight (100.9 ± 26.6 kg vs. 87.0 ± 25.0 kg; *p* < 0.001), body mass index (BMI) (34.3 ± 7.6 kg/m^2^ vs. 31.6 ± 9.9 kg/m^2^; *p* = 0.01), and hemodialysis duration before kidney transplant (5.6 ± 3.9 years vs. 4.1 ± 3.3 years; *p* < 0.001). Black patients were more likely to have end-stage renal disease due to hypertension (91 (72.8%) vs. 97 (53.9%); *p* = 0.001), although hypertension was the most common cause for end-stage renal disease in both groups.

Black patients were more likely to have a deceased donor kidney (117 (93.6%) vs. 133 (73.9%); *p* < 0.001). Cold ischemia time was similar between groups (13.5 ± 4.8 h vs. 14.3 ± 4.7; *p* = 0.341). A greater proportion of Black patients received induction with thymoglobulin (125 (100%) vs. 120 (66.7%); *p* < 0.001). Per protocol at the institution, all Black patients received thymoglobulin based on baseline rejection risks. There were similar rates of eculizumab usage between groups, which is used at our institution for ABO incompatible transplant (*p* = 0.704; Table 2). Black patients received significantly more propofol than non-Black patients intraoperatively (264.4 (193.2) mg vs. 193.8 (158.0) mg; *p* = 0.009). All other intraoperative variables were similar (Table 2).

Black patients had a longer duration of vasopressors (36.9 ± 66.8 h vs. 23.7 ± 31.7 h; *p* = 0.022) compared to non-Black patients (Table 3). Total use of all vasopressors in the intraoperative and postoperative periods in NEE was numerically but not statistically higher in Black patients (0.07 ± 0.1 NEE vs. 0.05 ± 0.1 NEE; *p* = 0.128). There was a higher dosage of intraoperative PE and postoperative NE in Black patients (*p* < 0.05; Table 3). Vasopressors were generally effective for both groups as each group had a similar number of instances of SBP < 120 mmHg (5.9 ± 6.0 vs. 7.1 ± 7.4; *p* = 0.134). The duration of time with SBP < 120 mmHg was numerically higher in Black patients, but this did not reach significance (Table 3). In bivariate correlation, only body weight, BMI, and baseline SBP were correlated with duration of vasopressor usage. Of note, race was not correlated with usage. A linear regression model was created (F = 9.550, df = 3, *p* < 0.001), and weight and SBP were associated with duration of vasopressor usage. For each kg of increased weight, there was an increased duration of vasopressor by 0.22 h (95% CI 0.031 to 0.414; *p* = 0.023). For each mmHg increase of baseline SBP, there was a decreased duration of vasopressor by 0.31 h (95% CI −0.458 to −0.162 h; *p* < 0.001).

Black patients were more likely to have worse allograft function with significantly higher rates of delayed graft function (49 [39.2%] vs. 42 [23.3%]; *p* = 0.003) and significantly lower rates of immediate graft function (49 [39.2%] vs. 98 [54.4%]; *p* = 0.010). Although there was a difference in the rates of hemodialysis requirement at postoperative day 7, there was no difference in hemodialysis requirement at day 28 (13 [10.2%] vs. 12 [6.7%]; *p* = 0.290). In bivariate correlation, race, weight, NEE, vasopressor duration, deceased donor kidney transplant (DDKT), and transplant surgery duration were correlated with the development of DGF. A logistic regression model was created (Chi-square = 59.444, df = 6, *p* < 0.001), and vasopressor duration and transplant surgery duration were associated with greater odds of developing DGF. Race was not correlated with DGF. For each hour of increased vasopressor duration, there was a 1.016 increased odds of DGF (95% CI 1.003 to 1.029; *p* = 0.013). For each hour of increased transplant surgery, there was a 1.556 increased odds of DGF (95% CI 1.080 to 2.243; *p* = 0.018).

There was no difference in safety outcomes between groups (Table 4). Rates of thrombosis were low, with only three events occurring, and only one event occurred while the patient was on AT2S. Atrial fibrillation was the most common arrhythmia to occur, with a similar incidence between groups (3 [2.4%] vs. 6 [3.3%]; *p* = 0.635).

## 4. Discussion

This retrospective chart review found that in patients receiving AT2S as first-line vasopressor for hypotension during the perioperative kidney transplant period, Black patients required significantly longer duration of vasopressors in total. There was no difference in the total amount of vasopressors between Black and non-Black patients. No difference was found for the amount of AT2S received between Black and non-Black patients, but there were differences in certain adrenergic agents. Race was the variable of interest in this study as physiological differences could affect response to vasopressors, but regression analysis in the study did not find that race was associated with vasopressor usage or worse allograft outcomes. Of interest, higher patient weights were correlated with longer duration on vasopressors, and consequently, longer duration on vasopressors was correlated with increased rates of delayed graft function.

This study explored the vasopressor effect in Black and non-Black patient populations because Black patients are known to have physiological differences in renin levels compared to White patients. We hypothesized that Black patients would have an exaggerated response to AT2S and potentially use less AT2S. This was not observed. Additionally, in the previous literature, Black patients have been observed to require higher doses and have a longer duration of adrenergic vasopressor therapy. We observed similar findings in our analysis.

The AT2S results may be different from our hypothesis, given our patient population. Post hoc analysis of the ATHOS-3 study found that patients with elevated renin levels above the study population median had lower rates of 28-day mortality [12]. Although Black patients should theoretically have elevated renin levels compared to the non-Black group in our study, we did not see any difference in outcomes other than vasopressor duration. Our institution serves a primarily Black and Hispanic population, so the non-Black group had a large percentage of Hispanic patients. The literature currently supports physiological differences that could affect vasopressor usage in Black patients compared to White patients, but it is unclear if there are differences in Black and Hispanic patients [10]. In patients with hypertension and not prescribed antihypertensives, Hispanic patients had higher plasma renin activity (PRA) compared to White and Black patients [13]. It is unclear in the ESRD, transplant, multi-antihypertensive agent, and shock populations how these profiles may differ, which could potentially confound our results. Potentially, utilizing RAAS profiling before transplant could provide additional data to inform vasopressor choice and expectations.

The duration of vasopressor usage and dosage of non-AT2S vasopressor results are similar to the previous literature, but with interesting differences observed in regression analysis. Although Black patients had a longer cumulative duration of vasopressors, regression analysis found that increased weight and not race was associated with increased duration of vasopressor use. Although many institutions do not perform kidney transplants on obese patients, this is not a barrier at our institution, as the average weight and BMI in Black patients were 100.9 kg versus 87 kg and 34.3 kg/m^2^ versus 31.6 kg/m^2^ in non-Black patients, respectively. AT2S was the only vasopressor that used weight-based dosing at our institution; all other vasopressors were dosed at non-weight-based rates. The difference in duration appears to be driven by the postoperative period. This could have been due to a difference in postoperative sedation or ventilator management, but these variables were not explicitly different in our investigation. There is a lack of evidence to support that weight is associated with duration of vasopressors, but it would be interesting to evaluate further, given the findings of this study [14].

The duration of vasopressors was associated with increased rates of delayed graft function. Although vasopressor duration was longer in Black patients, and Black patients were more likely to have DGF, our regression analysis did not find that race was independently associated with vasopressor usage or developing DGF. Given previous observational data, it is intuitive that longer vasopressor duration would be associated with increased rates of delayed graft function, especially if adrenergic agents are used [15]. Interestingly, it would be expected that race would be a variable associated with DGF, but in our analysis, this was not the case [16,17]. Further, although previous studies have shown that obesity is associated with delayed graft function, weight was not shown to be associated with delayed graft function through regression analysis in this study [18,19,20]. Perhaps immunosuppression management, the use of AT2S, or other unidentified practices mitigated these expected associations in our study. Regardless, given the association between duration of vasopressors and DGF, urgent investigation is warranted to minimize this vasopressor time post-kidney transplant.

There were several limitations to this study. First, there are differences in baseline variables that we attempted to control through regression analyses, but it is impossible to account for differences in comparative groups. Additionally, this study was retrospective in nature, with data captured via chart review. While we attempted to capture all data points, some variables were challenging to objectively collect and required extraction from chart notes, limiting the ability to objectively verify the data points. Lastly, this study was a comparison of Black to non-Black patients and assumes that our non-Black patients have similar physiological baselines to White patients in regard to their RAAS, which could potentially affect outcomes of AT2S use. This study was conducted at an institution with primarily Black and Hispanic patients, so the non-Black patient population was likely majority Hispanic patients, and the physiological differences of RAAS in Hispanic patients are not well defined.

## 5. Conclusions

Black patients had a longer duration of vasopressors, but this was not driven by differences in usage of AT2S. As baseline weight was significantly higher in Black patients and associated with duration of usage, perhaps the metabolic differences in our Black patients led to the observed differences. Regardless, longer duration of vasopressors was associated with delayed graft function, making this an area of utmost importance for continued investigation.

## Figures and Tables

**Figure 1 biomedicines-13-01819-f001:**
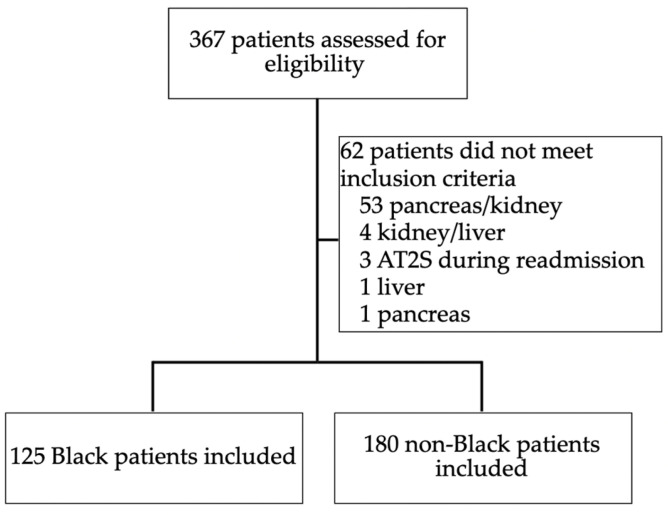
CONSORT diagram for evaluation of patients for inclusions and exclusions. AT2S = synthetic angiotensin II.

**Table 1 biomedicines-13-01819-t001:** Baseline characteristics.

	Black (n = 125)	Non-Black (n = 180)	*p*-Value
Age, years	53.4 ± 11.4	54.4 ± 12.6	0.482
Weight, kg	100.9 ± 26.0	87.0 ± 25.0	<0.001
Height, cm	171.2 ± 10.3	172.3 ± 75.5	0.870
BMI, kg/m^2^	34.3 ± 7.6	31.6 ± 9.9	0.010
Male	68 (54.4)	119 (66.1)	0.043
Hemodialysis Duration, years	5.6 ± 3.9	4.1 ± 3.3	<0.001
Past Medical History			
Hypertension	121 (96.8)	175 (97.2)	1.000
Hyperlipidemia	38 (30.4)	51 (28.3)	0.703
HFrEF	1 (0.8)	4 (2.2)	0.652
Atrial Fibrillation	9 (7.2)	20 (11.1)	0.322
Coronary Artery Disease	34 (27.2)	60 (33.3)	0.260
Type 2 Diabetes	56 (44.8)	93 (51.7)	0.247
ESRD Reason			
Hypertension	91 (72.8)	97 (53.9)	0.001
Type 2 Diabetes	49 (39.2)	74 (41.1)	0.813
Systemic Lupus Erythematosus	4 (3.2)	9 (5.0)	0.570
Focal Segmental Glomerulosclerosis	13 (10.4)	3 (1.7)	0.001
Unknown	2 (1.6)	22 (12.2)	<0.001
Baseline Vitals			
Systolic Blood Pressure, mmHg	140.5 ± 27.8	145.1 ± 27.6	0.156
Diastolic Blood Pressure, mmHg	72.5 ± 16.1	70.9 ± 16.1	0.405
Mean Arterial Pressure, mmHg	96.9 ± 20.8	96.8 ± 19.0	0.966
Heart Rate, bpm	85.7 ± 15.3	79.1 ± 14.1	<0.001
Respiratory Rate	18.3 ± 3.6	17.8 ± 3.2	0.187
Oxygen Saturation, %	97.8 ± 2.2	97.9 ± 2.0	0.606
Temperature, C	36.4 ± 0.5	36.4 ± 0.5	0.506

Categorical data is presented as n (%), and continuous data is presented as mean ± standard deviation unless otherwise noted. Abbreviations: BMI = body mass index, bpm = beats per minute, C = Celsius, HFrEF = heart failure reduced ejection fraction, kg = kilogram, m = meter, mmHg = millimeter of mercury, and % = percentage.

**Table 2 biomedicines-13-01819-t002:** Transplant characteristics and intraoperative variables.

	Black (n = 125)	Non-Black (n = 180)	*p*-Value
Donor Type			
Deceased	117 (93.6)	133 (73.9)	<0.001
Living Unrelated	3 (2.4)	25 (13.9)	<0.001
Living Related	5 (4.0)	22 (12.2)	0.014
Induction Agent			
Thymoglobulin	125 (100)	120 (66.7)	<0.001
Basilixumab	0 (0)	59 (32.8)	<0.001
Methylprednisolone	125 (100)	179 (99.4)	1.000
Eculizumab	2 (1.6)	5 (2.8)	0.704
Cold Ischemia Time, hours	13.5 ± 4.8	14.3 ± 4.7	0.341
Transplant Duration, hours	5.1 ± 1.0	5.0 ± 1.1	0.448
Donor Terminal Serum Creatinine, mg/dL	1.2 ± 0.8	1.4 ± 1.3	0.061
Fluids Received			
Albumin, mL	610.2 ± 232.3	595.2 ± 267.3	0.756
Mannitol, mL	192.0 ± 74.6	184.5 ± 123.9	0.578
Normal Saline, mL	864.3 ± 601.9	762.9 ± 487.1	0.693
Plasmalyte, mL	2480.5 ± 996.0	2358.4 ± 916.0	0.291
Lactated Ringers, mL	1062.8 ± 757.0	1163.2 ± 685.7	0.421
Packed Red Blood Cells, mL	549.0 ± 210.5	491.9 ± 240.3	0.185
Cryoprecipitate, mL	300.0 ± 212.1	333.3 ± 100.0	0.692
Intraoperative Ins, mL	3890.0 ± 1469.2	3891.5 ± 1253.4	0.543
Intraoperative Outs, mL ^T^	200 (0 to 380)	200 (0 to 650)	0.124
Intraoperative Sedatives Received			
Fentanyl, mcg	123.5 ± 45.9	125.0 ± 47.4	0.794
Hydromorphone, mg	0.60 ± 0.3	0.6 ± 0.3	0.598
Propofol, mg ^T^	227.4 (144.8 to 374.2)	164.1 (88.4 to 261.3)	0.010
Ketamine, mg	43.4 ± 16.6	46.2 ± 14.7	0.409
Postoperative Sedatives Received			
Propofol, mg	4409.8 ± 9375.7	5414.2 ± 6722.5	0.430
Ketamine, mg	409.8 ± 345.5	273.0 ± 207.8	0.489
Dexmedetomidine, mcg	3296.3 ± 7188.6	594.2 ± 961.0	0.174
% of Inhaled Anesthetic	1.6 ± 0.4	1.6 ± 0.3	0.090
% of Expired Anesthetic	1.4 ± 0.3	1.3 ± 0.3	0.083

Categorical data is presented as n (%), and continuous data is presented as mean ± standard deviation unless otherwise noted. Abbreviations: dL = deciliter, mcg = microgram, mg = milligram, mL = milliliter, and % = percentage. ^T^ Assessed with Mann–Whitney U test.

**Table 3 biomedicines-13-01819-t003:** Vasopressor and hemodynamic outcomes.

	Black (n = 125)	Non-Black (n = 180)	*p*-Value
Total Duration of Vasopressors, min	36.9 ± 66.8	23.7 ± 31.7	0.022
Intraoperative Duration of Vasopressors, min	1.9 ± 1.7	1.9 ± 1.8	0.848
Postoperative Duration of Vasopressors, min	35.0 ± 66.5	21.8 ± 31.3	0.21
Total NEE ^T^	0.07 ± 0.1	0.05 ± 0.1	0.128
Total Angiotensin II, ng/kg/min	15.2 ± 28.4	13.0 ± 16.5	0.384
Total Norepinephrine, mcg/kg/min	0.134 ± 0.115	0.098 ± 0.0807	0.393
Total Phenylephrine, mcg/kg/min	0.047 ± 0.047	0.045 ± 0.033	0.832
Total Vasopressin, units/min	0.182 ± 0.090	0.176 ± 0.126	0.946
Total Epinephrine, mcg/kg/min	0.111 ± 0.0665	0.072 ± 0.021	0.505
CIV Intraoperative Dose			
Angiotensin II, ng	215,079.38 ± 251,657.06	163,202.03 ± 268,943.87	0.076
Maximum Angiotensin II dose, ng/kg/min	18.9 ± 15.2	20.3 ± 12.4	0.487
Norepinephrine, mcg	495.75 ± 493.98	1027.33 ± 1570.02	0.269
Phenylephrine, mcg	7638.85 ± 6073.82	4371.25 ± 3248.85	0.020
CIV Postoperative Dose			
Angiotensin II, ng	28,010.07 ± 43,210.57	25,099.75 ± 28,241.09	0.280
Maxiumum Angiotensin II dose, ng/kg/min	24.3 ± 14.9	22.9 ± 14.8	0.479
Norepinephrine, mcg	38,034.4 ± 43,904.99	3002.58 ± 5541.68	0.015
Phenylephrine, mcg	469,585 ± 740,914.52	53,840 ± 52,334.82	0.068
# of SBP < 120 mmHg Deviations, hours	5.9 ± 6.0	7.1 ± 7.4	0.134
Duration of Time SBP < 120 mmHg, hours	10.0 ± 21.8	6.2 ± 12.9	0.056

Categorical data is presented as n (%), and continuous data is presented as mean ± standard deviation unless otherwise noted. Abbreviations: CIV = continuous intravenous infusion, kg = kilogram, mcg = microgram, min = minute, mmHg = millimeters of mercury, ng = nanogram, SBP = systolic blood pressure, and # = number. ^T^ Norepinephrine equivalents (NEEs) calculated according to [11].

**Table 4 biomedicines-13-01819-t004:** Secondary outcomes.

	Black (n = 125)	Non-Black (n = 180)	*p*-Value
Slow Graft Function	27 (21.6)	40 (22.2)	0.897
Immediate Graft Function	49 (39.2)	98 (54.4)	0.010
Delayed Graft Function	49 (39.2)	42 (23.3)	0.003
Hemodialysis at 28 Days	13 (10.2)	12 (6.7)	0.290
Mortality at 28 Days	2 (1.6)	0 (0)	0.167
Mechanical Ventilation	81 (64.8)	95 (52.8)	0.045
Thrombosis			
Deep Vein Thrombosis	1 (0.8)	2 (1.1)	0.787
Myocardial Infarction	0 (0)	1 (0.6)	0.404
New Arrhythmia on AT2S			
Atrial Fibrillation	3 (2.4)	6 (3.3)	0.635
Supraventricular Tachycardia	1 (0.8)	0 (0)	0.41
Ventricular Tachycardia	0 (0)	1 (0.6)	1

Categorical data is presented as n (%), and continuous data is presented as mean ± standard deviation unless otherwise noted. Abbreviations: AT2S = synthetic angiotensin II.

## Data Availability

Due to restrictions from our ethics office, data from this investigation are unavailable for distribution or sharing.

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
