# Peer review of "Comparison of Angiotensin II (Giapreza®) Use in Kidney Transplantation Between Black and Non-Black Patients"

_biomedicines, 2025, doi:10.3390/biomedicines13081819_

Round 1

Reviewer 1 Report

Comments and Suggestions for Authors

This is an interesting study comparing giapreza use in KT between black and non-black patients but i have some doubts.

Table 1 baseline values: how and when this value was obtained? It is one value only for each patient?

Table 1 past medical history: hypertension; hyperlipidemia, coronary artery disease should be defined

Table 2: any expanded criteria donor? why the use of eculizumb as induction agent? (in table 1 no patient had SHU as ESRD reason) 

Table 3: all patients received a vasopressor?

Author Response

Thank you for your review. Responses to your comments are in the word document attached. 

Reviewer 2 Report

Comments and Suggestions for Authors

Dear Editor and Authors,

I have reviewed this work before, I see that the given revisions were made meticulously. I do not have any new revision suggestions, thank you, with my respects,

Author Response

Thank you for your review. We have made edits to the methods/results section to more clearly express the research. 

On behalf of all our authors involved with this project, we sincerely thank you for your time and consideration in making thoughtful comments that will help us improve our final product. 

Reviewer 3 Report

Comments and Suggestions for Authors

The manuscript comparing the use of Angiotensin II (Giapreza®) for hypotension treatment in Black and non-Black kidney transplant patients poses an intriguing question. The findings are quite compelling and could provide valuable insights for managing hypotension in the Black population. Nonetheless, the authors acknowledge specific limitations in their research, particularly that the non-Black comparison group included individuals from the Hispanic population. This may indeed generate confusion and raise doubts about how the results are interpreted. Furthermore, the authors propose that the observed differences might stem from the metabolic traits of the patients (baseline weight was notably higher in Black patients, though it is important to note that both groups are obese, with elevated BMI), which again makes it more difficult to comprehend the data obtained. Nonetheless, in spite of potential limitations, this study holds importance as a possible reference for hospitals serving a large Black community, and publishing this work is definitely crucial, as it opens up avenues for further research and interpretation, ultimately leading to enhanced benefits for patients of all backgrounds.

Author Response

On behalf of all our authors involved with this project, we sincerely thank you for your time and consideration in making thoughtful comments that will help us improve our final product. 

Round 2

Reviewer 1 Report

Comments and Suggestions for Authors

Accept in present form